# Generalizable Imitation Learning from Observation via Inferring Goal Proximity

Youngwoon Lee[1][*][†]   Andrew Szot[2][*]   Shao-Hua Sun[1]   Joseph J. Lim[1][‡]

[1]University of Southern California   [2]Georgia Institute of Technology

https://clvrai.com/gpil

## Abstract

Task progress is intuitive and readily available task information that can guide an agent closer to the desired goal. Furthermore, a task progress estimator can generalize to new situations. From this intuition, we propose a simple yet effective imitation learning from observation method for a goal-directed task using a learned goal proximity function as a task progress estimator for better generalization to unseen states and goals. We obtain this goal proximity function from expert demonstrations and online agent experience, and then use the learned goal proximity as a dense reward for policy training. We demonstrate that our proposed method can robustly generalize compared to prior imitation learning methods on a set of goal-directed tasks in navigation, locomotion, and robotic manipulation, even with demonstrations that cover only a part of the states.

## 1 Introduction

Humans are effective at learning a task from demonstrations and applying the learned behaviors to other situations. We achieve this by extracting the underlying structure of the task when observing others fulfilling the task, instead of simply memorizing the demonstrator's low-level actions [4, 18]. This high-level task structure generalizes to new situations and thus helps us to quickly learn the task in new situations. One intuitive and readily available instance of such high-level task structure is *task progress*, measuring how much of the task the agent completed. Inspired by this insight, we propose a novel imitation learning method that utilizes task progress for better generalization to unseen states and goals.

Typical learning from demonstration (LfD) approaches [13, 35] greedily imitate the expert policy and thus suffer from accumulated errors causing a drift away from states seen in the demonstrations [38]. To make the imitation policy more robust to states not in demonstrations, adversarial imitation learning methods [14, 17] encourage the agent to stay near the expert trajectories using a learned reward that distinguishes expert and agent behaviors. However, such learned reward functions often overfit to the expert demonstrations by learning spurious correlations between task-irrelevant features and expert/agent labels [52], and thus suffer from generalization to slightly different initial and goal configurations from the ones seen in the demonstrations (e.g. holdout goal regions or larger perturbation in goal sampling).

To learn a more generalizable and informative reward from demonstrations, we propose an imitation learning from observation (LfO) method, which learns a task progress estimator and uses the task progress estimate as a dense reward for training a policy as illustrated in Figure 1. Unlike discriminating expert and agent behaviors by predicting *binary* labels in prior adversarial imitation

---

[*]Equal contribution. Correspondence to: `lee504@usc.edu` and `aszot3@gatech.edu`

[†]This work was partially carried out during an internship at NAVER AI Lab.

[‡]AI Advisor at NAVER AI Lab.

35th Conference on Neural Information Processing Systems (NeurIPS 2021).

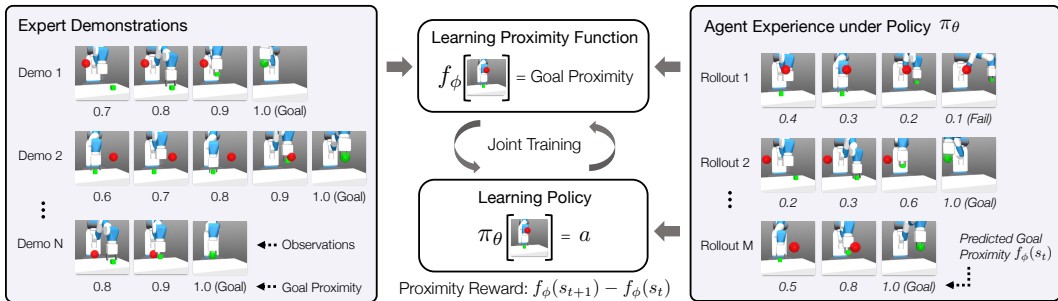

Figure 1: In goal-directed tasks, states on an expert trajectory have increasing proximity toward the goal as the expert makes progress towards fulfilling a task. Inspired by this intuition, we propose to learn a proximity function $f_\phi$ from expert demonstrations and agent experience, which predicts *goal proximity* (i.e. an estimate of temporal distance to the goal). Then, using this proximity function, we train a policy $\pi_\theta$ to progressively move to states with higher *predicted goal proximity (italicized numbers)* and eventually reach the goal. We alternate these two learning phases to improve both the proximity function and policy, leading to not only better generalization but also superior performance.

learning methods, which is prone to overfitting to task-irrelevant features, the task progress estimator is required to learn more task-relevant information to precisely predict the task progress on a *continuous* scale. Hence, it can generalize better to unseen states and provide more informative rewards.

As a measure of progress in goal-directed tasks, we define *goal proximity*, which is an estimate of temporal distance to the goal (i.e. the number of actions required to reach the goal) and entails all semantic information about how to reach the goal. We then train a *proximity function* to predict the goal proximity from expert demonstrations and agent experience. This proximity function acts as a dense reward to guide a reinforcement learning agent to reach states with high proximity, leading to the goal. In this paper, we focus on learning the proximity function and policy in a state space shared by the expert and learner, and leave generalizing to different embodiments as future work.

However, the predicted goal proximity can still be inaccurate on states not in the demonstrations, resulting in unstable policy learning. To improve the accuracy of the proximity function, we continually update it with trajectories from both the expert and learning agent. In addition, we penalize trajectories with the uncertainty of the proximity prediction to prevent the policy from exploiting inaccurate high proximity predictions. By leveraging the agent experience and predicting proximity function uncertainty, the proposed method achieves more efficient and stable policy learning.

The main contribution of this paper is an LfO algorithm for goal-directed tasks with better generalization to new goals or states not in demonstrations using goal proximity that informs an agent of the task progress. Together with a difference-based reward and uncertainty penalty of goal proximity estimation, our method provides more informative and robust rewards. Our extensive experiments show that the policy learned with the goal proximity function generalizes better than the state-of-the-art LfO algorithms on various goal-directed tasks, including navigation, locomotion, and robotic manipulation. Moreover, our method shows comparable results with LfD methods which learn from expert actions and a goal-conditioned imitation learning method which uses a sparse task reward.

## 2 Related Work

Imitation learning [39] aims to leverage expert demonstrations to acquire skills. While behavioral cloning [35] is simple but effective with a large number of demonstrations, it suffers from compounding errors caused by covariate shift [38]. On the other hand, inverse reinforcement learning (IRL) [1, 29, 51] estimates the underlying reward from demonstrations and trains a policy through reinforcement learning (RL) with this reward, which can better handle the compounding errors. Specifically, generative adversarial imitation learning (GAIL) [17] shows improved demonstration efficiency by training a discriminator to distinguish expert and agent transitions and using the discriminator output as a reward for policy training. GoalGAIL [9] further improves sample efficiency for goal-directed tasks by relabeling transitions [2] and using true environment rewards.

While these imitation learning algorithms require expert actions, imitation learning from observation (LfO) approaches learn from state-only demonstrations, such as videos and kinesthetic demonstrations. To imitate demonstrations without expert actions, inverse dynamics models [30, 33, 46], reachability functions [23], or learned reward functions [11, 26, 41, 42] can be learned and used for policy training, but training such models requires a large amount of quality data or additional test-time demonstrations. On the other hand, state-only adversarial imitation learning [47] can imitate from a few demonstrations.

However, in such adversarial imitation learning approaches, the discriminator tends to find spurious associations between task-irrelevant features and expert/agent labels [52]. This becomes problematic when the agent encounters unseen states and the discriminator erroneously assigns agent behaviors low scores based on these task-irrelevant features, providing a poor reward for the agent. To overcome finding spurious associations, in addition to discriminating expert and agent trajectories, we propose to also estimate the proximity to the goal, which requires more task-relevant information and thus generalizes better to new states.

Temporal progress estimation has shown its effectiveness as an auxiliary reward for RL [10, 24, 27] and decision making criteria [3, 6, 8]. However, these methods learn the progress estimator only from the given demonstrations. This hinders policy learning when the progress estimator fails to generalize to agent experience, allowing the agent to exploit inaccurate progress predictions for higher reward. Moreover, greedily choosing an action with the highest predicted temporal progress [3, 6, 8] could lead to low long-term returns. By incorporating online updates, uncertainty estimates, and a difference-based proximity reward, our method robustly learns from demonstrations to solve goal-directed tasks without access to expert actions or the true environment reward.

## 3 Method

In this paper, we address the problem of LfO for goal-directed tasks with a focus on generalization to states or goals not covered in the demonstrations. Adversarial LfO methods [47, 49] suggest learning a reward function that penalizes agent state transitions deviating from the expert trajectories. However, these learned reward functions often focus on task-irrelevant features [52] and do not generalize to states not in the demonstrations, leading to unsuccessful policy training.

To learn a generalizable reward, we propose to leverage task progress information freely available in demonstrations, in terms of *goal proximity*, which estimates temporal distance to the goal (i.e. number of actions required to reach the goal). Predicting precise goal proximity on a continuous scale, rather than simply distinguishing expert and agent states, requires the model to capture task-relevant information, allowing the proximity prediction to generalize to states not in the demonstrations (Section 3.2). Then, a policy learns to reach states with higher proximity prediction, leading to the goal (Section 3.3). Moreover, we propose to use the uncertainty of the proximity prediction to prevent the policy from exploiting over-optimistic proximity predictions and yielding undesired behaviors.

### 3.1 Preliminaries

We formulate our problem as a Markov decision process [44] defined through a tuple $(\mathcal{S}, \mathcal{A}, R, P, \rho_0, \gamma)$ of the state space $\mathcal{S}$, action space $\mathcal{A}$, reward function $R(s, a, s')$, transition distribution $P(s'|s, a)$, initial state distribution $\rho_0$, and discounting factor $\gamma$. We define a policy $\pi(a|s)$ that maps from a state $s$ to an action $a$ and correspondingly moves an agent to a new state $s'$ according to the transition probability $P(s'|s, a)$. The policy is trained to maximize the sum of discounted rewards, $\mathbb{E}_{(s_0, a_0, \ldots, s_{T_i}) \sim \pi} \left[ \sum_{t=0}^{T_i - 1} \gamma^t R(s_t, a_t, s_{t+1}) \right]$, where $T_i$ is the variable episode length.

In imitation learning, the learner receives a set of $N$ expert demonstrations, $\mathcal{D}^e = \{\tau_1^e, \ldots, \tau_N^e\}$. In this paper, we specifically consider the LfO setup where each demonstration $\tau_i^e$ is a sequence of states. Moreover, we assume that goal information is explicitly or implicitly included in the state $s$, and all demonstrations are successful; therefore, the final state of each trajectory achieves the task goal.

### 3.2 Learning Goal Proximity Function

To effectively leverage expert demonstrations and generalize to new states or new goals, learning a generalizable reward function is essential. In goal-directed tasks, an estimate of how close an

agent is to the goal can be utilized as a dense and direct learning signal. Moreover, predicting the *continuous* goal proximity requires understanding the task structure and thus encourages finding more task-relevant features, resulting in better generalization.

Therefore, instead of learning to simply discriminate agent and expert trajectories, we propose to learn a *goal proximity function*, $f : \mathcal{S} \rightarrow [0, 1]$, which predicts *goal proximity* of a state $s$, which is a discounted value based on the temporal distance to the goal (i.e. inversely proportional to the number of actions required to reach the goal). In this paper, we define goal proximity as the exponentially discounted proximity $f(s_t) = \delta^{(T_i - t)}$, where $\delta \in (0, 1)$ is a discounting factor and $T_i$ is the episode length. Note that the goal proximity function measures the temporal distance, not the spatial distance, between the current and goal states. Therefore, a single proximity value can entail all information about the task, goal, and any roadblocks. There are alternative ways to define goal proximity, such as linearly discounted proximity [24] and ranking-based proximity [5, 6]. But, in this paper, we use the exponentially discounted proximity as it performs better across most tasks (see appendix, Figure 8).

We train a goal proximity function $f_\phi$ parameterized by $\phi$ to minimize the following objective:

$$\mathcal{L}_\phi = \mathbb{E}_{\tau_i^e \sim \mathcal{D}^e, s_t \sim \tau_i^e} \big[ f_\phi(s_t) - \delta^{(T_i - t)} \big]^2. \tag{1}$$

Since the goal proximity function trained only on expert demonstrations can overfit to the data, we further train the goal proximity function with online agent experience by setting the target proximity of states in agent trajectories to 0, similar to adversarial imitation learning methods [17]:

$$\mathcal{L}_\phi = \mathbb{E}_{\tau_i^e \sim \mathcal{D}^e, s_t \sim \tau_i^e} \big[ f_\phi(s_t) - \delta^{(T_i - t)} \big]^2 + \mathbb{E}_{\tau \sim \pi_\theta, s_t \sim \tau} \big[ f_\phi(s_t) \big]^2. \tag{2}$$

By learning to predict the goal proximity, $f_\phi$ not only learns to discriminate agent and expert trajectories (i.e. predict 0 proximity for an agent trajectory and positive proximity for an expert trajectory) but also acquires the task information about temporal progress entailed in the trajectories. From this freely available additional supervision, the proximity function is required to learn task-relevant features. Hence, the resulting proximity function generalizes better to unseen states and provides more informative learning signals to the policy as empirically shown in Section 4.

Due to the lack of environment reward, *successful* agent experience is also used as negative examples for proximity function training, and thus the proximity function learns to predict low goal proximity even for successful trajectories. However, early stopping and learning rate decay can ease this problem [52], and the optimal proximity function still outputs the average of expert and agent labels, which is $\delta^{(T_i - t)}/2$ for ours and 0.5 for GAIL [14].

### 3.3 Training Policy with Proximity Reward

In a goal-directed task, a policy $\pi_\theta$ aims to get close to and eventually reach the goal. We can formalize this objective as maximizing the difference-based *proximity reward* $R_\phi$, the increase in goal proximity, at every timestep, which corresponds to making consistent progress towards the goal:

$$R_\phi(s_t, s_{t+1}) = f_\phi(s_{t+1}) - f_\phi(s_t). \tag{3}$$

Given the proximity reward $R_\phi$, the policy is trained to maximize the expected discounted return:

$$\mathbb{E}_{(s_0, \ldots, s_{T_i}) \sim \pi_\theta} \Bigg[ \sum_{t=0}^{T_i - 1} \gamma^t R_\phi(s_t, s_{t+1}) \Bigg]. \tag{4}$$

However, a policy trained with the proximity reward can sometimes acquire undesired behaviors by exploiting over-optimistic proximity predictions on states not seen in the expert demonstrations. This becomes critical when the expert demonstrations are limited and cannot sufficiently cover the state space. To avoid inaccurate predictions leading an agent to undesired states, we propose to (1) fine-tune the proximity function with online agent experience to reduce optimistic proximity predictions; and (2) penalize agent trajectories with high uncertainty in goal proximity prediction.

To alleviate the effect of inaccurate proximity estimation in policy training, we discourage the policy from visiting states with uncertain proximity estimates. Specifically, we model the uncertainty $U_\phi(s_t)$ as the disagreement of an ensemble of proximity functions by computing the standard deviation of their outputs [22, 31]. Then, we use this estimated uncertainty to penalize exploration of states with

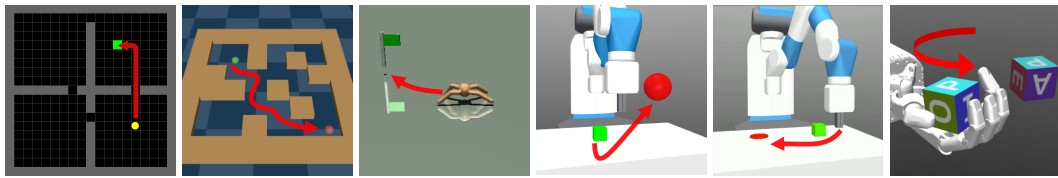

(a) NAVIGATION (b) MAZE2D (c) ANT REACH (d) FETCH PICK (e) FETCH PUSH (f) HAND ROTATE

Figure 2: Six goal-directed tasks are used for our experiments. (a) The agent must navigate across rooms to reach the goal. (b) The agent needs to navigate the maze to reach the goal. (c) The ant agent must walk towards the flag. (d, e) The robotic arm is required to pick up or push the block towards the goal (red). (f) The dexterous robot hand needs to rotate the block in-hand to the desired rotation.

high uncertainty. The proximity estimate $f_\phi(s_t)$ is the average prediction of the ensemble. With the uncertainty penalty, the modified proximity reward can be written as:

$$R_\phi(s_t, s_{t+1}) = f_\phi(s_{t+1}) - f_\phi(s_t) - \lambda \cdot U_\phi(s_{t+1}), \qquad (5)$$

where $\lambda$ is a tunable hyperparameter to balance the proximity reward and uncertainty penalty. A larger $\lambda$ results in more conservative exploration outside the states covered by the expert demonstrations.

In summary, we propose to learn a goal proximity function to robustly provide a reward signal on states or goals not covered by demonstrations. We train the goal proximity function to estimate how close the current state is to the goal, and train a policy to maximize the goal proximity while avoiding states with uncertain proximity predictions. We jointly train the proximity function and policy as described in appendix, Algorithm 1.

## 4 Experiments

In this paper, we propose a generalizable LfO algorithm that leverages task progress information (i.e. goal proximity) freely acquired from demonstrations. Hence, in our experiments, we aim to answer the following questions: (1) Does our method lead to policies that generalize better to states and goals not in the demonstrations? (2) How does our method's efficiency and performance compare against prior work in LfO and LfD? (3) What factors contribute to the performance of our method? To answer these questions we consider diverse goal-directed tasks: navigation, locomotion, and robotic manipulation.

### 4.1 Experimental Setup

To demonstrate the improved generalization capabilities of policies trained with the goal proximity, we benchmark our method under two different setups: expert demonstrations are collected from (1) only a fraction of the possible initial and goal states (e.g. 25%, 50% coverage) and (2) initial states with smaller amounts of noise. These generalization experimental setups serve to mimic the reality that expert demonstrations may be collected in a different setting from agent learning. For instance, due to the cost of demonstration collection, the demonstrations may poorly cover the state space, which corresponds to the setup (1). Likewise, in the setup (2), demonstrations may be collected in controlled circumstances with little noise. Then, an agent in an actual environment would encounter more noise than presented in the demonstrations, leading to a wider initial state distribution.

In our experiments, we use the discounting factor $\delta = 0.95$ for the goal proximity. We use an ensemble of 5 proximity functions to model uncertainty across all tasks. For policy optimization, we use PPO [40], which is widely used in LfO and LfD methods, and its hyperparameters are tuned for each method and task (see appendix, Table 2). Each baseline implementation is verified against the results reported in its original paper. We train each task with 5 random seeds and report mean and standard deviation. See Section F for further implementation details.

### 4.2 Baselines

We compare our method to the state-of-the-art methods in LfO (BCO, GAIfO, GAIfO-s) as well as LfO with reward (GoalGAIL) and LfD (BC, GAIL, SQIL) approaches, which require additional supervisions, such as task reward and expert actions:

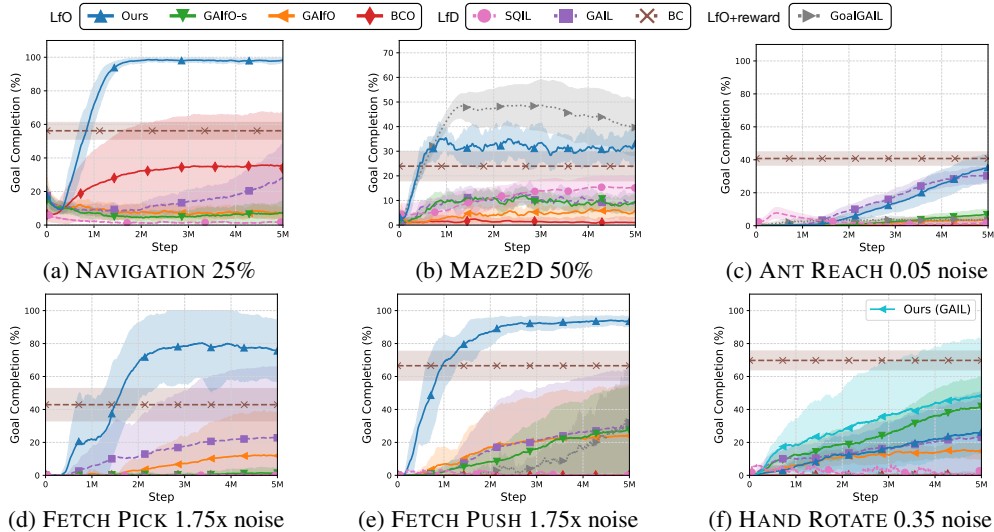

Figure 3: Goal completion rates of our method and baselines. The agent must generalize to a wider state and goal distributions than seen in the demonstrations. Demonstrations cover only a part of states (a, b) or are generated with less noise (c, d, e, f). Our method learns more stably, faster, and achieves higher goal completion rates than prior LfO methods. Moreover, our method outperforms the LfD baselines in NAVIGATION, FETCH tasks, and MAZE2D, and achieves comparable results in ANT REACH. GoalGAIL performs well in MAZE2D since it can easily acquire environment rewards.

- **BCO** [46] learns an inverse dynamics model from environment interaction to provide action labels in demonstrations for behavioral cloning.
- **GAIfO** [47] trains a discriminator with state transitions $(s, s')$, instead of $(s, a)$ as in GAIL.
- **GAIfO-s** [49] learns a discriminator based off a single state, not a state transition as with GAIfO.
- **GoalGAIL** [9] uses goal reaching reward and relabeling to improve sample efficiency of GAIL.
- **BC** [35] fits a policy to the demonstration state-action pairs $(s, a)$ with supervised learning.
- **GAIL** [17] is an adversarial imitation learning with a discriminator trained on state-action pairs $(s, a)$ from both expert and agent.
- **SQIL** [37] is a sample-efficient imitation learning method which adds expert transitions $(s, a)$ with reward 1 to the replay buffer of off-policy RL and assigns 0 reward to all agent experience.

## 4.3 Navigation

We first examine the NAVIGATION task between four rooms shown in Figure 2a to demonstrate generalization capability of our method, and visualize the learned goal proximity function. The agent observes the $19 \times 19 \times 4$ 2D map of the maze and moves in one of four directions. In this task, the agent starting and goal positions are randomly sampled (see an example in appendix, Figure 12). We provide 250 expert demonstrations obtained using a shortest path algorithm. During demonstration collection, we hold out 0%, 25%, 50%, and 75% of the possible agent starting and goal positions uniformly at random. In contrast, during agent learning and evaluation, start and goal positions are sampled from all possible positions.

Figure 3a shows that our method achieves near 100% success rate in 2M environment steps even with demonstrations only covering 25% of starting and goal states, while other LfO methods fail to learn the task. Although BC, GAIL, and BCO achieve success rates of about 60%, 30%, and 35%, respectively, they show limited generalization to unseen configurations. This result shows that the learned goal proximity function generalizes well to unseen configurations.

Figure 4d visualizes the proximity function trained with 50% coverage demonstrations and 250k steps of agent training. Our proximity function predicts high proximity near the goal and lower proximity when the agent is farther away from the goal. This demonstrates that our proximity function can learn

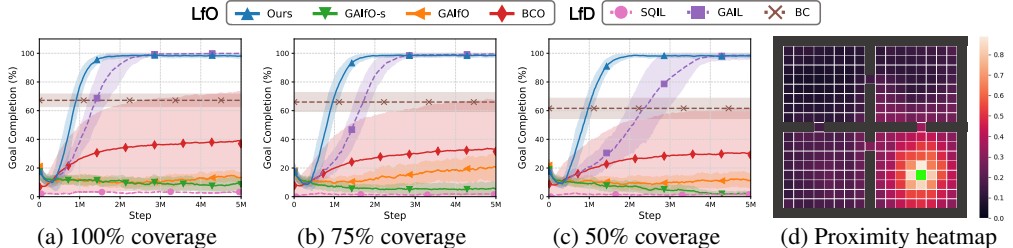

Figure 4: Analyzing the effect of improved generalization as the cause for performance increase in our method. (a) Performance with no generalization required. (b, c) Performance with increasing difference between start and goal distributions of demonstrations and agent learning. (d) Visualization of the learned proximity function for a fixed goal (green) in the 50% coverage case. The proximity function was evaluated for every state on the grid; lighter cells correspond to states with higher estimated proximity to the goal.

the semantic, non-euclidean relationship between high-dimensional observations and goals. Since the proximity function is conditioned on the state, similar states are likely to have similar predicted proximity, and thus the proximity function learns a spatially consistent measure of proximity from temporal supervision. Moreover, as the task progress is a relative position within a trajectory, both slow and fast demonstrations result in the same task progress. More visualizations can be found in appendix, Section E.

Finally, we investigate our hypothesis that the goal proximity function allows for greater generalization, which results in better performance with smaller demonstration coverages. We compare the cases where extreme (25% coverage), moderate (50% and 75% coverage), and no generalization (100% coverage) are required. Figure 3a and Figure 4 show that our method consistently achieves almost 100% success rates in 2M steps across all coverages and is not as affected by the increasingly difficult generalization settings as baselines. In contrast, all LfO baselines struggle to learn the task when the demonstrations do not cover all configurations. LfD methods also shows limited generalization in 25% coverage since the discriminator can easily learn spurious associations between the actions and labels, which hurts generalization to new actions. This supports our hypothesis that the goal proximity function is able to capture the task structure and therefore, generalize better to unseen configurations.

## 4.4 Maze2D

We further evaluate our method in MAZE2D [15] with the medium maze, a continuous version of NAVIGATION. The agent observes its position, velocity, and goal position, and then outputs an x- and y-velocity to navigate the maze. The agent starting and goal positions are randomly sampled. We collect 100 demonstrations (18k transitions) using a planner from Fu et al. [15].

Our method outperforms LfO baselines over all demonstration coverages (see appendix, Figure 7). More importantly, in the low coverage case, our method outperforms BC, which has access to expert actions, as shown in Figure 3b. This could be because our proximity function generalizes well whereas BC is not robust to unseen states under small demonstration coverages. On the other hand, GoalGAIL shows the best performance regardless of coverages as the task can be easily solved with the sparse reward and goal relabeling, which is not available for our method and other baselines.

## 4.5 Ant Locomotion

In ANT REACH [16], the quadruped ant is tasked to reach a randomly generated goal, which is along the perimeter of a half circle of radius $5\,\mathrm{m}$ centered around the ant (see Figure 2c). The 132D state consists of joint angle, velocity, contact force, and the goal position relative to the agent. We collect 1k demonstrations (25k transitions) using the pre-trained policy (trained for 40M steps). When demonstrations are collected, no noise is added to the initial pose of the ant whereas random noise is added to the initial pose during policy learning, which requires the reward functions to generalize to unseen states.

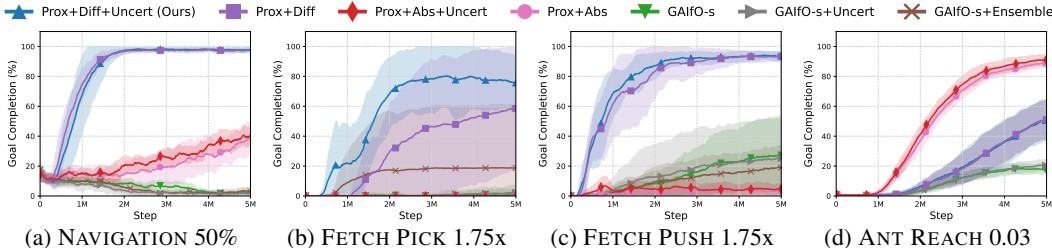

(a) Navigation 50%  (b) Fetch Pick 1.75x  (c) Fetch Push 1.75x  (d) Ant Reach 0.03

Figure 5: Analysis of the contribution of goal proximity function, uncertainty penalty, and reward formulation to the performance. "Prox" uses the goal proximity function while "GAIfO-s" does not. "+Diff" uses $R(s_t, s_{t+1}) = f(s_{t+1}) - f(s_t)$ and "+Abs" uses $R(s_t) = f(s_t)$ as a reward. "+Uncert" adds the uncertainty penalty to the reward. "+Ensemble" uses an ensemble for the discriminator.

In Figure 3c, with 0.05 added noise, our method achieves 35% success rate while BCO, GAIfO, and GAIfO-s achieve 1%, 2%, and 7%, respectively. This result illustrates the importance of learning proximity as opposed to discriminating expert and agent states for generalization to unseen states. The performance of GAIfO and GAIfO-s drops drastically with larger joint angle randomness as shown in appendix, Figure 7. As ANT REACH is not as sensitive to noise in actions compared to other tasks, BC and GAIL show superior results but our method still achieves comparable performance.

## 4.6 Robotic Manipulation

We evaluate our method in two robotic manipulation tasks with the 7-DoF Fetch robotics arm: FETCH PICK and FETCH PUSH [34]. The robot must grasp and move a block to a target position for FETCH PICK, and push a block to a target position for FETCH PUSH. The 16D state consists of the gripper pose, object pose, the gripper pose relative to the object, and goal position. Both the initial and target positions of the block are randomly initialized. We generate 1k demonstrations using a hard-coded policy, consisting of 33k and 28k transitions for FETCH PICK and FETCH PUSH, respectively. The policy is trained in an environment with larger noise applied to the starting and target block positions.

In FETCH PICK, our method achieves about 80% success rate outperforming all baselines, despite LfD methods learning with expert actions (see Figure 3d). The best performing baseline BC only obtains around 40% success rate. The high variance in performance between seeds comes from the difficulty of learning the grasping behavior with large noise. In FETCH PUSH, our method outperforms baselines in generalization to unseen states by achieving more than 90% success rate (see Figure 3e). This shows that our proximity function is able to accelerate policy learning in continuous control environments with superior generalization capability.

## 4.7 Dexterous Hand Manipulation

We evaluate our method in a challenging in-hand object manipulation task [34], HAND ROTATE as shown in Figure 2f. In HAND ROTATE, a 24-DoF Shadow Dexterous Hand must in-hand rotate a block to a target $z$-axis rotation. The state consists of the agent's joint angles and velocities, object pose, and the target rotation. Due to the high dimension of the state (68D) and action space (20D), HAND ROTATE is extremely challenging for both RL and IL without dense reward. We therefore ease the task by constraining the possible initial and target $z$ rotations to $\left[-\frac{\pi}{32}, \frac{\pi}{32}\right]$ and $\left[\frac{\pi}{3}, \frac{\pi}{2}\right]$. We collect 10k demonstrations (98k transitions) using a pre-trained policy (trained for 8M steps).

In Figure 3f, GAIfO-s performs well because its reward function is biased to provide large negative rewards encouraging the agent to end the episode early which is only possible by succeeding. In contrast, our difference-based reward is designed to provide positive rewards, which does not exploit this task property, and performs poorly even with an additional constant penalty -0.005 every step. To test the generalization capability of our proximity function, we additionally examine a variant of our method (Ours-GAIL), which uses the same reward formulation as GAIfO-s, $\log f_\phi(s_t) - \log(1 - f_\phi(s_t))$. With this biased reward function, our method outperforms both GAIfO-s and GAIL, which verifies the benefit of our proximity function in generalization to noisy environments. While BC achieves the high success rate with 10x more demonstrations compared to other tasks, SQIL shows poor performance due to the lack of the negative reward bias.

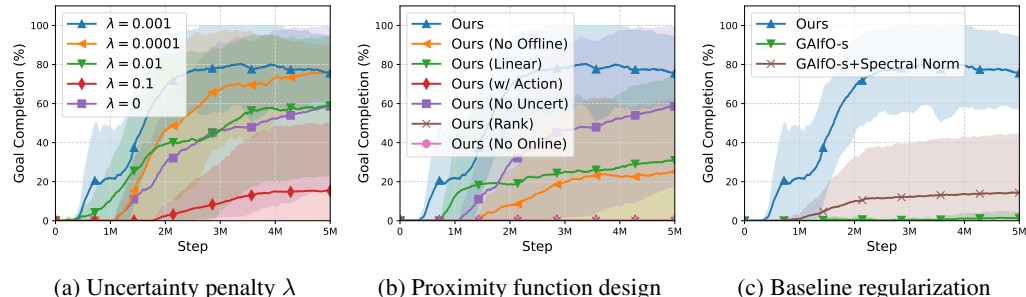

| (a) Uncertainty penalty $\lambda$ | (b) Proximity function design | (c) Baseline regularization |

Figure 6: Ablation of our method and comparison to a regularized baseline on FETCH PICK 1.75x to investigate (a) effects of uncertainty penalty coefficient $\lambda$; (b) effects of proximity function design and online/offline training; and (c) generalization capability of a regularization technique and our proximity function.

## 4.8 Ablation Study

**Dissecting proximity reward**   We analyze the contribution of the proximity function, reward formulation, and uncertainty penalty to our method's performance across four tasks in Figure 5. Adding uncertainty to GAIfO-s (GAIfO-s+Uncert) produced a 18.4% boost in average success rate compared to regular GAIfO-s. Proximity supervision, without the uncertainty penalty, resulted in a 66.7% increase in average performance over GAIfO-s with the difference-based reward $f(s_{t+1}) - f(s_t)$ (Prox+Diff) and 25.8% with the absolute reward $f(s_t)$ (Prox+Abs). This higher performance means **modeling proximity is more important than the uncertainty penalty** for our method.

Although we choose difference-based reward with exponentially discounted goal proximity, the goal proximity can be either linear or exponential discounting, and both can be used for either a difference-based or absolute reward, which perform differently across tasks. For example, the difference-based proximity reward is better for policy learning than the absolute proximity reward except on ANT REACH and HAND ROTATE, where the bias of the absolute reward [20] helps the agent survive longer and reach the goal. This is a fundamental problem in IRL, where inductive bias in reward functions is crucial and varies by tasks [20]. Nonetheless, our extensive experiments (Figure 5, 6b, 8) show that our goal proximity reward provides a more stable and generalizable learning signal than baselines under the same reward bias.

Moreover, we found that **the uncertainty penalty and proximity function have a synergistic interaction**. Combining both the proximity and uncertainty gives a 68.7% increase with the difference-based reward (Prox+Diff+Uncert) and 26.4% increase with the absolute reward (Prox+Abs+Uncert). The uncertainty penalty is especially important for the proximity function as it models fine-grain temporal information where inaccuracies can be easily exploited, as opposed to the binary classification of other adversarial imitation learning methods.

**Ensemble networks**   Next, we study if the robustness of our method comes from the use of ensemble networks or task progress. We verify this by applying ensemble of discriminators to the best performing baseline, GAIfO-s. Figure 5 shows that GAIfO-s with ensemble networks (GAIfO-s+Ensemble) only achieves 19.6% higher success rates, but this is still 39.7% lower than our method on average. Therefore, **the use of task progress is key** to learn a generalizable reward, not the use of ensemble networks.

**Regularization of discriminators**   In our experiments above, we show that our goal proximity function is generalizable to unseen states and goals, which leads to successful imitation learning. We verify whether standard regularization techniques, such as spectral normalization [28], can also provide the same generalization benefit. In the FETCH PICK 1.75x noise setting (Figure 6c), GAIfO-s without regularization struggles to learn, achieving only a 1.43% success rate. Not surprisingly, applying spectral normalization [28] to the discriminator of GAIfO-s improves the success rate to 14.56%, which suggests that generalization of the reward function is key to imitation learning with insufficient demonstration coverage. Despite this improvement, our method performs much better at 75.45% success. In summary, **predicting goal proximity enables significantly better**

**generalization than regularization** on the baselines. Figure 10 in appendix show similar results across most other tasks.

**Uncertainty penalty coefficient** $\lambda$    In Figure 6a, we investigate how the uncertainty penalty coefficient $\lambda$ affects the performance, showing that our method performs the best with $\lambda = 0.001$. A higher or lower $\lambda$ yields worse performance since a higher $\lambda$ prevents exploring unseen states while a lower $\lambda$ encourages exploiting uncertain predictions.

**Proximity function training**    In Figure 6b, we test the importance of online and offline training of the proximity function. Note that we update the policy with online interactions in both scenarios. The result shows that **online proximity function update is crucial** for our method as the agent fails without online update. Meanwhile, no pre-training, Ours (No Offline), slows down training. Similar results can be observed across all tasks (see appendix, Figure 8).

Our ablation experiments show that (1) goal proximity generalizes better and is more informative for policy learning; (2) the difference-based proximity reward generally performs better than the absolute one; and (3) the uncertainty penalty boosts the performance of our method. In conclusion, all three components of proximity, difference-based reward, and uncertainty are crucial for our method.

# 5    Conclusion

We propose a generalizable learning from observation (LfO) method inspired by how humans acquire generalizable task information and learn skills in new situations by watching others performing goal-directed tasks. We specifically propose to use task progress, which is intuitive and readily available task information that can guide an agent closer to the goal. Inspired by this insight, we learn a goal proximity function and utilize it as a dense reward for policy learning. We hypothesize that predicting the task progress requires more task-relevant information than estimating an occupancy measure [17], and thus generalizes to states not seen in the demonstrations. Our extensive experiments on navigation, locomotion, and robotic manipulation show that our goal proximity function improves generalization in imitation learning, which results in better performance compared to LfO methods and comparable performance with LfD methods which learn from expert actions.

In imitation learning, the *generalization* ability can include generalization to (1) unseen states and goals, (2) new visual environments (e.g. background), (3) unseen objects, and (4) different embodiments (e.g. humans to robots or different dynamics). In this paper, we focus on generalization to (1) unseen states and goals. This is especially important in imitation learning when the number of demonstrations is not sufficient to cover all possible states and goals. This is very common in imitation learning due to costly demonstration collection. Our approach suggests an effective way of using the demonstrations with limited coverage by learning the generalizable goal proximity reward.

Generalization to a different environment and embodiment is another important research direction and this is indeed our immediate future work. Recent advances in generalizable representation learning [42, 43, 48], robust policy learning [19, 21], and cross-domain correspondence [50] enable us to train a policy that generalizes to new environments and embodiments. Yet, these approaches are orthogonal and complementary to our method as our goal proximity function can be trained on top of the learned representations [19, 21, 42, 43, 48, 50]. We believe that our method can be combined with these approaches and improve their performance with better demonstration efficiency and additional supervision about task progress.

**Societal Impact**    Our method aims to increase the ability of autonomous agents, such as robots and self-driving cars, to imitate experts (e.g. humans) from observation alone. This enables autonomous agents to utilize data even without expert actions, such as kinesthetic demonstrations and video demonstrations. Ultimately, it could allow autonomous agents to acquire skills even from watching Youtube videos. Since our method learns from experts, it inherits any biases of the demonstrator, such as sub-optimal or unsafe behaviors. Additionally, demonstrations are an easy and intuitive way to specify behaviors, its potential for automation is a threat to job security. However, we overall see enormous benefit with this technology increasing human quality of life and automating difficult jobs.

## Acknowledgments and Disclosure of Funding

This research is partially supported by the Annenberg Fellowship from USC and the funding from National Science Foundation (NSF NRI-2024768) and NAVER AI Lab. We thank members of the USC CLVR lab for constructive feedback.

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
