## A    Comparison with GAIL and Its Variants

Our method shares a similar adversarial training process with GAIL [17]. First of all, similar to the discriminator in GAIfO-s [49], our proximity function takes only the current state as input. However, rather than training the discriminator to classify expert from agent, we train the proximity function to regress to the proximity labels which are 0 for agent and the time discounted value between 0 and 1 for expert. Our reward formulation also differs from GAIL approaches which gives a log probability reward based on the discriminator output. We instead incorporate a proximity estimation uncertainty penalty and a difference-based proximity reward as shown in Equation 3.

## B    Failure of GAIfO and SQIL

We found that SQIL training is unstable and often collapses after some amount of training steps (see Ant experiments in Figure 7). Similar trends can be observed in the original paper [37] and other recent papers [36, 45]. We hypothesize that GAIfO easily overfits to demonstrations compared to other baselines (e.g. GAIfO-s) since GAIfO conditions its discriminator on both the current and next observations.

We evaluated these methods with the demonstrations from the same initial and goal state distributions in the first column of Figure 7. Even though they are trained for the same goal distributions as the demonstrations, they still overfit to the demonstration states and thus cannot generalize to unseen states encountered during online rollouts for most tasks.

## C    Analysis on Generalization of Our Method and Baselines

By learning to predict the goal proximity, the proximity function not only learns to discriminate expert and agent states but also models task progress, which encourages acquiring task-relevant information. With this additional supervision on learning goal proximity, we expect the proximity function to provide a more informative learning signal to the policy and generalize better to unseen states than baselines which easily overfit the reward function to expert demonstrations. To analyze how well our method and the baselines can generalize to unseen states, we vary the difference between the states encountered in expert demonstrations and agent training as described in Section 4.

One way we vary the difference between expert demonstrations and agent learning is restricting the expert demonstrations to only cover parts of the state space. For NAVIGATION and MAZE2D, we show results for expert demonstrations that cover 100%, 75%, 50%, and 25% of the state space. For the discrete state space in NAVIGATION, we restrict expert demonstrations to the fraction of possible agent start and goal configurations. For MAZE2D, we break the maze into $6 \times 6$ cells and sample a part of the cells for starting states and another part for goal states.

Likewise, we also measure generalization by adding more noise to the initial state during agent learning. On FETCH PICK, FETCH PUSH, ANT REACH, and HAND ROTATE we show results for four different noise settings. For the two FETCH tasks, the 2D sampling region of the object and goal is scaled by the noise factor. For ANT REACH, uniform noise scaled by the noise factor is added to the initial joint angles, whereas the demonstrations have no noise. For HAND ROTATE, uniform noise scaled by the noise factor is added to the possible initial and target object pose. If our method allows for greater generalization from the expert demonstrations, our method should perform well even under states different than those in the expert demonstrations.

The results of our method and baselines across varying degrees of generalization are shown in Figure 7. Note that the results in the main paper are for 1.75x noise in FETCH PICK and FETCH PUSH, 0.05 noise in ANT REACH, 0.35 noise in HAND REACH, and 25% coverage in Maze2D. Across both harder and easier generalization, our method demonstrates more consistent performance compared to baseline methods. While GAIfO-s performs well on high coverage or low noise, which require little generalization in agent learning, its performance deteriorates as the expert demonstration coverage decreases.

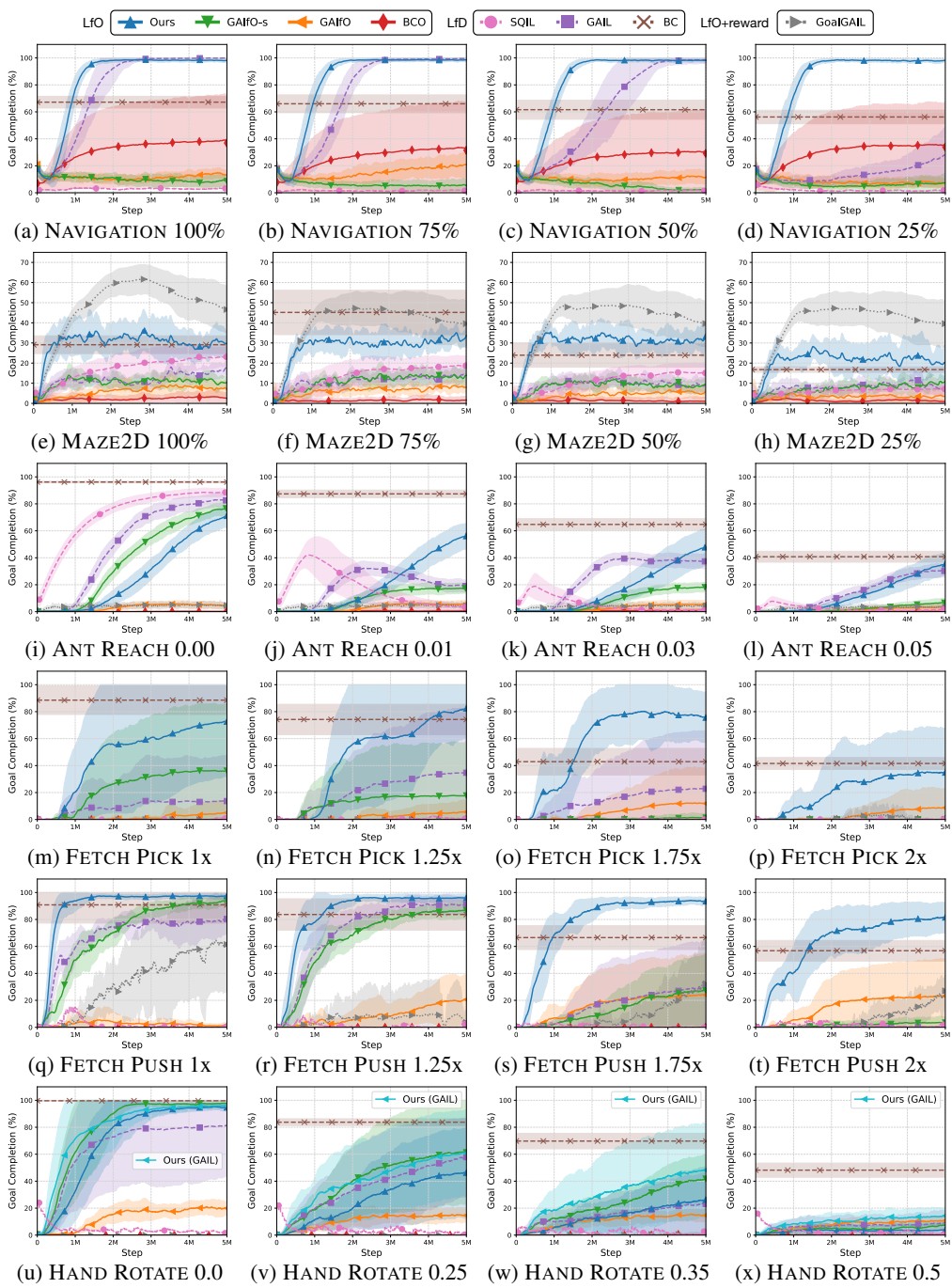

Figure 7: Analyzing generalization to unseen states from expert demonstrations. NAVIGATION and MAZE2D tasks are tested with different coverages of state spaces in demonstrations, while FETCH, ANT REACH, and HAND ROTATE tasks are tested in more noisy environments. The number indicates the amount of additional noise in agent learning compared to that in the expert demonstrations, with more noise requiring harder generalization. The noise level increases from left to right.

# D  Further Ablations

We include additional ablations to further highlight the advantages of our main proposed method over its variants. We evaluate against the same ablations proposed in the main paper (Figure 6b), but across all environments. We present all these results in Figure 8.

Our method shows the best performance in the majority of environments. In all tasks, incorporating online updates is crucial since the proximity function can overfit to expert trajectories and poorly generalize to agent trajectories. Updating the proximity function with online agent experience lowers the proximity prediction outside of the expert trajectories and thus leads an agent to follow the expert. Our method with the uncertainty penalty shows superior performance in FETCH PICK and HAND ROTATE, while it performs similarly with our method without the uncertainty penalty in other environments. Our method using the linear proximity function achieves similar to or slightly lower performance than the exponential proximity function used in the main paper. Offline pre-training of the proximity function is also helpful in most environments.

We also compare to an ablation which learns the proximity function through a ranking-based loss similar to Brown et al. [5], Burke et al. [6]. However, we empirically found it to be ineffective and difficult to train. This ranking-based loss uses the criterion that for two states from an expert trajectory $s_{t_1}, s_{t_2}$, the proximities should obey $f(s_{t_1}) < f(s_{t_2})$ if $t_1 < t_2$. We therefore train the proximity function with the cross entropy loss $-\sum_{t_i < t_j} \log \frac{\exp f_\phi(s_{t_j})}{\exp f_\phi(s_{t_i}) + \exp f_\phi(s_{t_j})}$. We incorporate agent experience by adding an additional loss which ranks expert states above agent states for randomly sampled pairs of expert and agent states $(s_e, s_a)$ through the cross-entropy loss:

$$-\sum_{s_a \sim D^e, s_e \sim \pi_\theta} \log \frac{\exp f_\phi(s_e)}{\exp f_\phi(s_a) + \exp f_\phi(s_e)}. \tag{6}$$

Unlike the discounting factor in the discounting-based proximity function, the ranking-based training requires no hyperparameters. However, as shown in Figure 8, the lack of supervision on ground truth proximity scores results in less meaningful predicted proximity and a worse learning signal for the agent, which could explain its poor performance.

We also show results for applying spectral normalization [28] to GAIfO-s [47] in Figure 10 across all tasks. While regularizing the GAIfO-s discriminator can consistently improve its performance, it still cannot generalize as well as our method for the majority of tasks. As mentioned in Section 4.7,

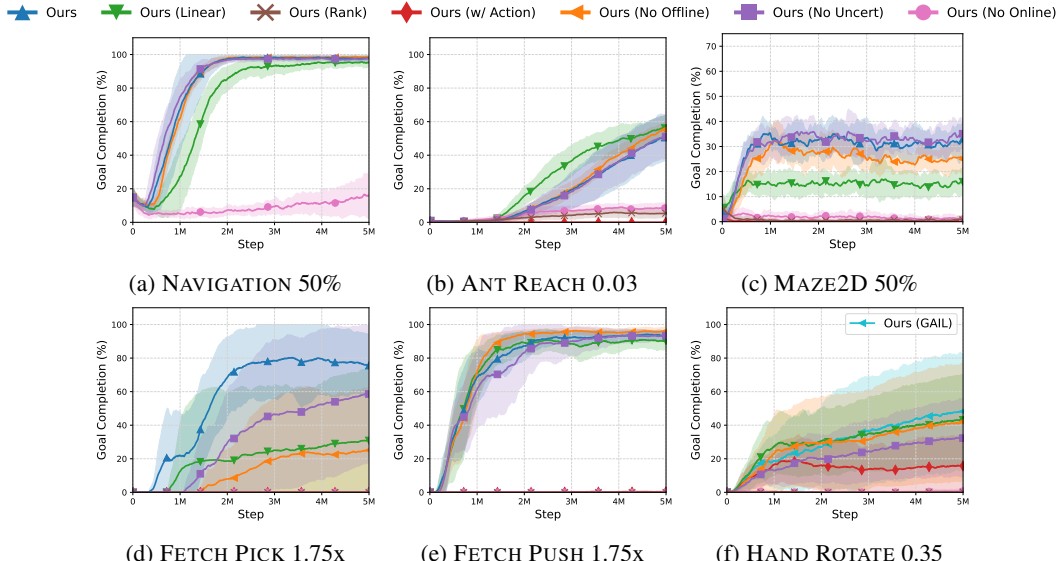

Figure 8: Ablation on proximity function design and online/offline proximity function training. We compare our method to the proximity function with actions as input or with a ranking-based objective (Equation 6). Our method shows consistently superior or comparable performance over all ablations.

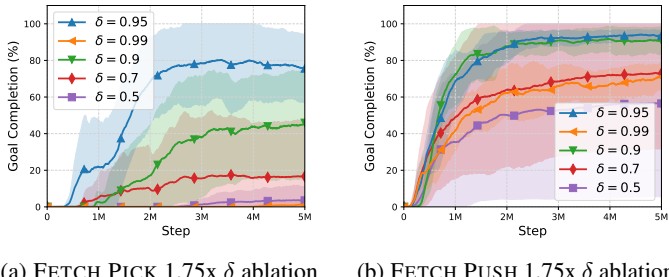

(a) FETCH PICK 1.75x $\delta$ ablation          (b) FETCH PUSH 1.75x $\delta$ ablation

Figure 9: Analyzing different choices of the proximity discounting factor $\delta$ for training the proximity function. The model learns similarly well over a range of $\delta$ values around 0.95, but struggles for too large or too small $\delta$.

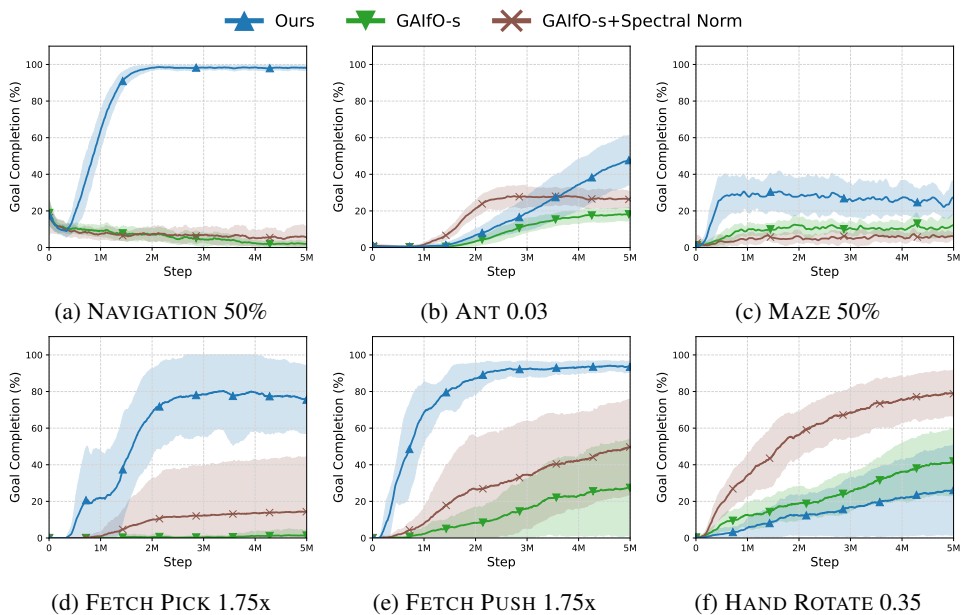

(a) NAVIGATION 50%          (b) ANT 0.03          (c) MAZE 50%

(d) FETCH PICK 1.75x          (e) FETCH PUSH 1.75x          (f) HAND ROTATE 0.35

Figure 10: Effect of applying spectral normalization to the GAIfO-s baseline compared to the performance of our method. While regularization helps GAIfO-s, it still is outperformed by our method in the majority of tasks.

GAIfO-s has a bias to provide negative rewards encouraging the agent to end the episode early, which is a desirable property for the HAND ROTATE task. Vanilla GAIfO-s therefore performs better than our method in this environment, and spectral normalization for the discriminator further improves GAIfO-s performance.

# E   Qualitative Results

It is important for agent learning that the proximity function gives higher values for states that are temporally closer to the goal. To verify this intuition, we visualize the proximity values predicted by the proximity function in a successful episode from agent learning in Figure 11. In Figure 11, we can observe that the predicted proximity increases as the agent moves closer to the goal (except HAND ROTATE). This provides an example of the proximity function generalizing to agent experience and providing a meaningful reward signal for agent learning.

We notice that while the predictions increase as the agent nears the goal, the proximity prediction values are often low (<0.1) as shown in Figure 11c. These low values are mostly predicted for the states not covered in the demonstrations due to the adversarial online training of the proximity

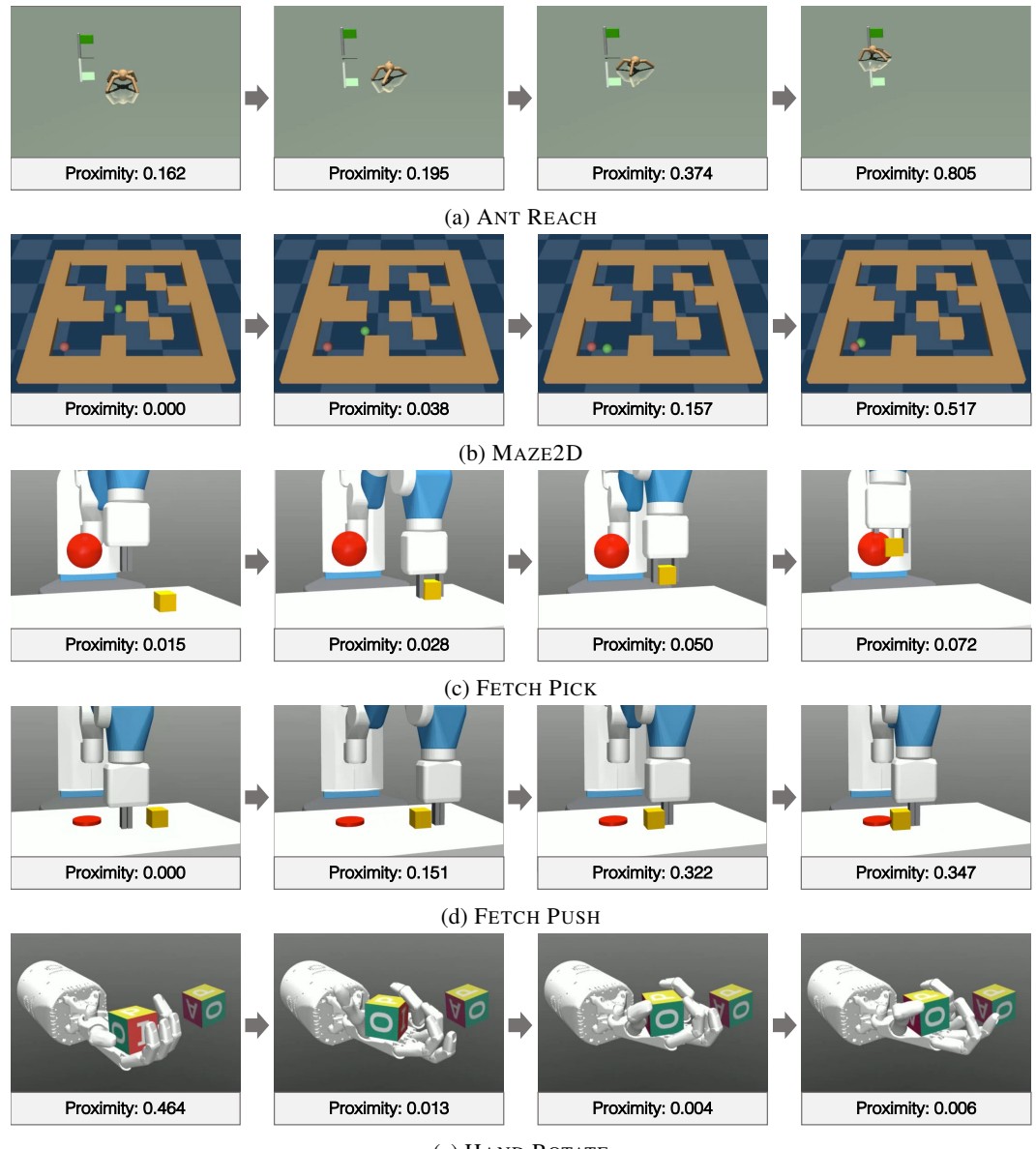

(a) ANT REACH

(b) MAZE2D

(c) FETCH PICK

(d) FETCH PUSH

(e) HAND ROTATE

Figure 11: Visualizing the proximity predictions for a successful trajectory from agent learning. Four informative frames are selected from the overall trajectory and the predicted proximity value is displayed below. The proximity prediction visualization for NAVIGATION can be found in Figure 4d.

function. During online proximity function training, we label agent experience with 0 proximity and therefore proximity predictions get lower, especially for states not in the demonstrations.

For HAND ROTATE, the proximity function fails to predict increasing proximity for states near the goal as an agent cannot learn to imitate the exact expert trajectories. Instead, due to the negatively biased reward, the agent finds a new way to solve the task as discussed in Section 4.7 and therefore achieves low proximity predictions even for successful trajectories as shown in Figure 11e. However, our method still provides relatively higher proximity values near goal states compared to baseline methods, which leads our agent to achieve higher performance in noisy environments.

# F   Implementation Details

We use PyTorch [32] for our implementation and all experiments are conducted on a workstation with an Intel Xeon E5-2640 v4 CPU and a NVIDIA Titan Xp GPU. Most adversarial imitation learning methods and our method are trained for around 3 hours with 32 parallel workers. GoalGAIL and SQIL training takes around 48 hours since they use off-policy optimization with a single worker.

## F.1   Environment Details

In this section, we summarize details of the six goal-directed tasks discussed in this paper. For all environments, the starting and goal states are randomly initialized. All units in this section are in meters and radians unless otherwise specified. The summary of observation spaces, action spaces, and episode lengths are described in Table 1. To evaluate the generalization capability of our method and baselines, we constrain the coverage of expert demonstrations or add additional starting state noise during agent learning as discussed in Section C.

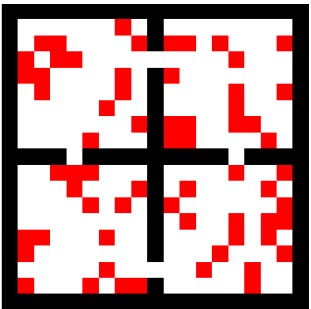

Figure 12: The goals of the expert demonstrations in red for the NAVIGATION 25% holdout setting.

**NAVIGATION [7]**   In NAVIGATION, the state consists of a one-hot vector for each grid cell encoding wall, empty space, agent, or goal. NAVIGATION has four discrete actions for moving in four directions. We collect 250 expert demonstrations using the shortest path algorithm, BFS search. The 25% holdout region is visualized in Figure 12.

**MAZE2D [15]**   In MAZE2D, the state consists of the agent's 2D position, velocity, and the goal position. The point mass agent moves around the maze by controlling the continuous value of its $(x, y)$ velocity. The only modification in this environment from `maze2d-medium-v1` [15] is the episode length reduced from 600 to 400. We collect 100 expert demonstrations using a planner provided by Fu et al. [15].

**ANT REACH [16]**   In ANT REACH the state consists of joint angle, velocity, force and the relative goal position, and the agent is controlled using joint torque control. We collect 1,000 demonstrations using an expert policy trained using PPO [40] based on the reward function $R(s, a) = 1 - 0.2 \cdot ||p_{ant} - p_{goal}||_2 - 0.005 \cdot ||a||_2^2$, where $p_{ant}$ and $p_{goal}$ are $(x, y)$-positions of the ant and goal, respectively, and $a$ is an action. Please refer to the code for more details.

**FETCH PICK and FETCH PUSH [34]**   The actions in the FETCH experiments use 3-D end-effector position control and 1-D continuous control for the gripper (fixed for FETCH PUSH). A 16-dimensional state in FETCH tasks consists of the relative position of the goal from the object, relative position of the end-effector to the object, and robot joint state. We found that not including the velocity information was beneficial for all learning from observation approaches in FETCH tasks. In FETCH PICK, we generate 1,000 demonstrations by hard coding the Sawyer Robot to first reach above the object, then reach down and grasp, and finally move to the target position. Similarly, in FETCH PUSH, we collect 664 demonstrations by hard coding the Sawyer to reach behind the object and then execute a planar push towards to the goal.

**HAND ROTATE [34]**   The original task `HandManipulateBlockRotateZ-v0` proposed in Plappert et al. [34] is challenging to solve without reward due to its large and combinatorial state space and large action space. Hence, we reduce the initial and goal $z$ rotations of the block to $[-\frac{\pi}{32}, \frac{\pi}{32}]$ and $[\frac{\pi}{3}, \frac{\pi}{2}]$. The 68-D state space consists of the agent's joint angles and velocities, and object pose. The 20-D action space is for joint torque control of 24-DoF Shadow Dexterous Hand. We collect 10,000 demonstrations using an expert policy trained with DDPG+HER [2] using a sparse reward.

## F.2   Network Architectures

**Actor and critic networks:** We use the same architecture for actor and critic networks except for the output layer where the actor network outputs an action distribution while the critic network outputs a critic value. For NAVIGATION, the actor and critic network consists of $CONV(3, 2, 16) - ReLU -$

$MaxPool(2,2) - CONV(3,2,32) - ReLU - CONV(3,2,64)$ followed by two fully-connected layers with hidden layer size 64, where $CONV(k,s,c)$ represents a $c$-channel convolutional layer with kernel size $k$ and stride $s$. For other tasks, we model the actor and critic networks as two separate 3-layer MLPs with hidden layer size 256. For the continuous control tasks, the last layer of the actor MLP has two heads to output the mean and standard deviation of a Gaussian distribution which an action is sampled from. We use the ReLU activation for NAVIGATION and $\tanh$ for other tasks.

**Goal proximity function and discriminator:** The goal proximity function and discriminator use a CNN encoder (the same CNN architecture as the actor and critic networks) followed by a hidden layer of size 64 for NAVIGATION and a 3-layer MLP with a hidden layer of size 64 for other tasks. When measuring the uncertainty of predictions, we use an ensemble of 5 networks.

### F.3  Training Details

For our method and all baselines except BC [35] and BCO [46], we train policies using PPO [40]. The hyperparameters for policy training are shown in Table 2, while the hyperparameters for the proximity and discriminator function are shown in Table 3. For our method, we found it helpful to normalize the reward based on the moving average and standard deviation of returns. We also did so for baselines when it helped.

For hyperparameter tuning, we searched over entropy coefficients $\{0.0001, 0.001, 0.01\}$, state normalization $\{\text{True}, \text{False}\}$, uncertainty coefficient $\{0.0001, 0.001, 0.01, 0.1\}$, learning rates $\{0.0001, 0.0003, 0.001\}$, and reward normalization $\{\text{True}, \text{False}\}$.

In BC, the demonstrations were split into 80% training data and 20% validation data. The policy was trained on the training data until the validation loss stopped decreasing. The policy is then evaluated for 1,000 episodes to get an average success rate.

In GAIfO-s and GAIL, we use the reward form of $\log D(s) - \log(1 - D(s))$ and $\log D(s,a) - \log(1 - D(s,a))$, respectively, from Finn et al. [12], Fu et al. [14].

For GoalGAIL [9], we use the default hyperparameters used in the original implementation. For the policy network, we use a deterministic policy for DDPG [25] and use the $\tanh$ activation to normalize the policy output between $[-1, 1]$. We update the policy and critic every 2 environment steps and the discriminator every 10 environment steps to prevent overfitting.

---

**Algorithm 1** Imitation learning with learned goal proximity

---

**Require:** Expert demonstrations $\mathcal{D}^e = \{\tau_1^e, \ldots, \tau_N^e\}$
 1: Initialize goal proximity function $f_\phi$ and policy $\pi_\theta$
 2: **for** $i = 0, 1, ..., M$ **do**
 3:     Sample expert demonstration $\tau^e \sim \mathcal{D}^e$
 4:     Update $f_\phi$ with $\tau^e$ to minimize Equation 1            ▷ Offline proximity function training
 5: **end for**
 6: **for** $i = 0, 1, ..., L$ **do**
 7:     Rollout trajectories $\tau_i = (s_0, \ldots, s_{T_i})$ with $\pi_\theta$
 8:     Compute proximity reward $R_\phi(s_t, s_{t+1})$ for $(s_t, s_{t+1}) \sim \tau_i$ using Equation 5
 9:     Update $\pi_\theta$ using any RL algorithm                        ▷ Policy training
10:     Update $f_\phi$ with $\tau_i$ and $\tau^e \sim \mathcal{D}^e$ to minimize Equation 2     ▷ Online proximity function training
11: **end for**

---

Table 1: Environment details. In NAVIGATION and MAZE2D, the goal and agent are randomly initialized anywhere on the grid. In ANT REACH, the angle of the goal and the velocity of the agent are randomly initialized. The goal and object noise in FETCH describes the amount of uniform noise applied to the $(x, y)$ coordinates of the object and goal. In HAND ROTATE, the state and goal noises are applied to the initial and goal object rotations.

| | State | Action | Goal noise | State noise | Episode len. | # demos |
|---|---|---|---|---|---|---|
| NAVIGATION | (19, 19, 4) | 4 | - | - | 50 | 250 |
| MAZE2D | 6 | 2 | - | - | 600 | 100 |
| ANT REACH | 132 | 8 | $\theta \in [0, \pi]$ | $v \in [\pm.005]$ | 50 | 1,000 |
| FETCH PICK | 16 | 4 | $(x, y) \in [\pm.02, \pm.05]$ | | 50 | 1,000 |
| FETCH PUSH | 16 | 3 | $(x, y) \in [\pm.02, \pm.05]$ | | 60 | 664 |
| HAND ROTATE | 68 | 20 | $\theta \in [\frac{\pi}{2}, \frac{\pi}{3}]$ | $\theta \in [\pm\frac{\pi}{32}]$ | 50 | 10,000 |

Table 2: PPO hyperparameters used for baselines and our method.

| Hyperparameter | Value |
|---|---|
| Learning Rate | 3e-4 |
| Learning Rate Decay | Linear decay |
| # Mini-batches | 4 (NAVIGATION), 32 (others) |
| # Epochs per Update | 4 (NAVIGATION), 10 (others) |
| Discount Factor $\gamma$ | 0.99 |
| Rollout Size | 16,000 (ANT REACH), 4,096 (others) |
| Entropy Coefficient | 0.01 (NAVIGATION), 0.001 (others) |
| State Normalization | False (NAVIGATION), True (others) |

Table 3: Hyperparameters for goal proximity functions (ours) and discriminators (baselines).

| Hyperparameter | Value |
|---|---|
| # Networks for Ensemble | 5 |
| # Epochs for Pre-training | 5 |
| Discount Factor $\delta$ | 0.95 (exponential), $1/H$ (linear) |
| Uncertainty Coefficient $\lambda$ | 0.001 (FETCH), 0.01 (others) |
| Learning Rate (ours) | 1e-3 (NAVIGATION, FETCH, MAZE2D), 1e-4 (ANT REACH, HAND ROTATE) |
| Learning Rate (baselines) | 1e-4 |
| Batch Size | 32 (NAVIGATION), 128 (others) |
| # Updates per Agent Update | 1 |
| Experience Buffer Size | 16,000 (ANT REACH), 4,096 (others) |
| Reward Norm. (ours) | True |
| Reward Norm. (baselines) | True (FETCH), False (others) |