# OpenReview forum: "Generalizable Imitation Learning from Observation via Inferring Goal Proximity"
_NeurIPS.cc/2021/Conference — NeurIPS 2021 Poster_

### Official Review · Reviewer_YZMg · 2021-07-12

**Rating:** 7
**Confidence:** 5

**Summary:**

Disclaimer:
I was the reviewer of this paper's older version submitted to ICML 2021. Along the reviewing process, the authors' information was kept anonymous.

Overall I'm glad to see this improved version. As the authors stated, 1) comparison to Goal-GAIl added; 2) clearly specify the generalization is limited, and not referring to out-of-demo goals/states which including a different embodiment (for example unseen objects) of environments. Thanks for the authors' improved clarity.

This paper is well written with sound evaluation results, falling in approaches learning reward function from human demonstrations, though I prefer viewing the generalization problem of adapting human demonstrations to unseen states or actions as more of a representation learning or meta-learning problem since such problem settings will flexibly apply to the manipulation of unseen objects, which is important to learn generalizable manipulation skills.

**Limitations And Societal Impact:**

Yes, it was discussed in the last paragraph of Page 9.

**Main Review:**

Learning a robust reward function from human demonstrations is an old topic studied in literature like IRL, GAIL, etc. To overcome overfitting to the human demonstrations, typically, online agent experience is used to continuously improve the reward function estimation. This paper utilizes the same approach by introducing a goal proximity function in a common adversarial training scenario.

The goal proximity function estimates how many discounted time steps are still needed to reach the target. After that, the reward function is simply the proximity difference between successive states. As a result, two inductive biases are assumed here:
- (1). the goal proximity, i.e., how much time steps are left to reach the target.
- (2). The task reward is simply a function of S_{t} and S_{t+1} that is independent of robot actions.

Major comments:

- (1). While I have the impression that the above two assumptions are too strong in many RL tasks in general, the evaluation results presented in the six tasks convince me that the proposed method will still apply well in a variety of tasks.
- (2). The claimed generalization from human demonstration seems only mean to generalize to different initial and target configurations in all the tasks. And in the Conclusion section, the authors discussed the possibility of generalizing to tasks with different task settings, for example, objects or backgrounds, but leaving such generalizations as future works. This actually limit the proposed method's applicability since such generalization is now widely studied in approaches leveraging both expert demonstration data and agent online training samples.


**Time Spent Reviewing:**

2 hours

---

> ### Author Response · Authors · 2021-08-10
> **Response to Reviewer YZMg**
>
> We thank the reviewer for the constructive feedback and address the concerns in detail below.
>
>
> **1. Two strong assumptions required: (1) task progress can be computed as the number of steps left to the goal and (2) the reward function only depends on states not actions.**
>
> As the reviewer pointed out, our method leverages the inductive bias of goal proximity and state-conditioned reward function. Optimal demonstrations by definition satisfy the assumption (1) that the task progress is the number of steps left to the goal since these demonstrations follow an optimal path to the goal. The learning from observation problem formulation naturally requires the assumption (2) since actions are not available in the expert demonstrations. As the reviewer mentioned, our extensive experiments suggest that our method can be applied to a variety of tasks despite these assumptions.
>
>
> **2. Generalization in this paper is limited to unseen states and goals.**
>
> As described in conclusion, the _generalization_ ability in imitation learning can include generalization to (1) unseen states and goals, (2) new visual environments (e.g. background), (3) unseen objects, and (4) different embodiments (e.g. humans to robots or different dynamics).
>
> In this paper, we focus on generalization to (1) unseen states and goals. This is especially important in imitation learning when the number of demonstrations is not sufficient to cover all possible states and goals. This is very common in imitation learning due to costly demonstration collection. Our approach suggests an effective way of using the demonstrations with limited coverage by learning the generalizable goal proximity reward.
>
> We agree that generalization to a different environment and embodiment is another important research direction and this is indeed our immediate future work. As Reviewer YZMg mentioned, recent advances in generalizable representation learning [1-3], robust policy learning [4-5], and cross-domain correspondence [6-7] enable us to train a policy that generalizes to new environments and embodiments. Note that these approaches are orthogonal and complementary to our method as our goal proximity function can be trained on top of the learned representations [1-7]. We believe that our method can be combined with these approaches and improve their performance with better demonstration efficiency and additional supervision about task progress. We will include this discussion in the revised version.
>
>
> **References**
>
> [1] Sermanet et al. “Time-Contrastive Networks: Self-Supervised Learning from Video”, ICRA 2018
>
> [2] Srinivas et al. “Universal Planning Networks”, ICML 2018
>
> [3] Wang et al. “NerveNet: Learning Structured Policy with Graph Neural Networks”, ICLR 2018
>
> [4] Kumar et al. “RMA: Rapid Motor Adaptation for Legged Robots”, RSS 2021
>
> [5] James et al. “Transferring End-to-End Visuomotor Control from Simulation to Real World for a Multi-Stage Task”, CoRL 2017
>
> [6] Zhang et al. “Learning Cross-Domain Correspondence for Control with Dynamics Cycle-Consistency”, ICLR 2021
>
> [7] Zhang et al. “Policy Transfer across Visual and Dynamics Domain Gaps via Iterative Grounding”, RSS 2021

---

> > ### Comment · Reviewer_YZMg · 2021-08-16
> > **Re: Response to Reviewer YZMg**
> >
> > Thanks for your comments. Well explained!

---

### Official Review · Reviewer_AFg8 · 2021-07-17

**Rating:** 7
**Confidence:** 4

**Summary:**

This paper presents a learning from observation technique for solving goal-conditioned tasks from demonstrations and environment interaction, without access to environment provided rewards. The method takes an adversarial learning approach like prior works, where the agent tries to learn behavior that is indistinguishable from that of the demonstrator. However the key difference in this work is that instead of using a binary discriminative objective, the paper uses a temporal progress prediction objective, where the learning agent’s experience is labeled as having 0 temporal progress and the demonstrators experience is densely labeled based on progress in the episode. The learning RL agent then aims to maximize progress (while also avoiding high uncertainty regions)

The intuition is such an objective will provide (1) a denser reward signal and (2) generalize more effectively to unseen states/goals. The paper demonstrates over a range of domains including navigation, locomotion, and manipulation this seems to hold true, with improved stability and performance over most baselines when generalizing to unseen states.

**Main Review:**

*Strengths*

Overall the paper presents a simple and effective idea.
- Since the data comes from a demonstrator, it is reasonable to assume that the distances are near optimal, and thus that learning a dense measure of temporal task progress from the demos would be effective.
- It also is reasonable that such an objective will force the discriminator to pay attention to features which measure progress throughout the task, and thus may be able to generalize to new states and goals more effectively than the binary 0/1 adversarial objectives used in prior works.
- The experiments are thorough, comparing against the relevant baselines in GAIfO and SQIL. They show that across several task domains the when generalizing to unseen states and goals performance using this method is much stronger than the baselines. Figure 4 also nicely shows how GAILs performance degrades as the more and more states are held out while the proposed method does not.
- The ablations also suggest that the different components of the method, including the uncertainty penalty and using the difference in score between states is important.

*Weaknesses*

There are a few results in the experiments which could use more explanation:
- First, I found it surprising how poorly SQIL and GaifO performed across all environments, failing out outperform even vanilla BC.  Is the main failure mode simply that these methods cannot effectively handle unseen states? I think that could make sense, but it would be valuable to confirm then that if the evaluation states/goals are the same as what exist in the demos then these methods should perform well. Figure 4a seems to be testing this exactly, yet GAIfO and SQIL still fail completely. Why do they fail in this setting?
- Second, I was confused by the difference in performance between using the absolute proximity vs the difference in proximity between time-steps. On 3/4 tasks it seems using the absolute fails completely, and on the 4th using the absolute significantly outperforms using the difference. Could the authors explain further why this difference exists, and which types tasks benefit which approach?
- Figure 6a seems to indicate that the methods performance is actually quite sensitive to the weighting of the uncertainty parameter, so not sure the paper can claim “our method is robust to different choices of the uncertainty penalty coefficient λ“.

**Time Spent Reviewing:**

2.5

---

> ### Author Response · Authors · 2021-08-10
> **Response to Reviewer AFg8**
>
> We thank the reviewer for the constructive feedback and address the concerns in detail below.
>
>
> **1. Why do SQIL and GAIfO fail completely?**
>
> We found that SQIL training is unstable and often collapses after some amount of training steps (see Fig 7i-7l). Similar trends can be observed in the original paper [1] and other recent papers [2, 3]. Moreover, we hypothesize that GAIfO easily overfits to demonstrations compared to other baselines (e.g. GAIfO-s) since GAIfO conditions its discriminator on both the current and next observations.
>
> As Reviewer AFg8 suggested, we evaluated these methods with the demonstrations from the same initial and goal state distributions in the first column of Fig. 7. Even though they are trained for the same goal distributions as the demonstrations, they still overfit to the demonstration states and thus cannot generalize to unseen states encountered during online rollouts for most tasks. We will include this discussion about the failures of SQIL and GAIfO in the revised version.
>
>
> **2. Why does the absolute proximity reward work so well in Ant Reach?**
>
> In general tasks with the quadruped ant, the agent should first learn to stand up and move stably to acquire any meaningful reward signals; otherwise, it terminates right away without any reward signal. Due to this strong inductive bias on survival, the absolute proximity reward, which directly encourages the agent to stay alive, significantly improves the results of the Ant Reach task. In contrast, in most other tasks, this inductive bias is not as critical as in Ant Reach, and hence the difference-based reward guides the agent to the goal more effectively. This finding is described in lines 305-307.
>
>
> **3. In Figure 6a, the choice of $\lambda$ parameter seems to be important.**
>
> We agree that $\lambda$ is an important parameter for our method and we thank the reviewer for pointing this out. As suggested by Reviewer AFg8, we will clarify this in the revised version.
>
>
> **References**
>
> [1] Reddy et al. “SQIL: Imitation Learning via Reinforcement Learning with Sparse Rewards”, ICLR 2020
>
> [2] Rafailov et al. “Visual Adversarial Imitation Learning using Variational Models”, arXiv 2021
>
> [3] Swamy et al. “Of Moments and Matching: A Game-Theoretic Framework for Closing the Imitation Gap”, ICML 2021

---

> > ### Comment · Reviewer_AFg8 · 2021-08-16
> > **Re: Response**
> >
> > Thanks to the authors for the detailed response:
> >
> > Re 1 - Got it that makes sense, it would be good to add more discussion of this in the paper. As a side note - if the issue is in the discriminator overfitting, there are a number of simple tricks that have been used in adversarial imitation works in the past to address this, like mixup regularization, spectral normalization, and data augmentation. It might even be worth playing around with these and seeing if including these has an impact on the results - for example, with regularization does the performance gap between SQIL and the proposed method shrink?
> >
> > Re 2 - Got it ok. Thinking about the total un-discounted finite-horizon return, the difference would be that with absolute reward the total return is the sum over all time steps, while with the difference based reward it would just be the final state reward - the initial state reward. And so the optimal policy under the different reward functions would actually be different, with the former encouraging solving the task more quickly. Perhaps this explains the difference?

---

> > > ### Author Response · Authors · 2021-08-18
> > > **Response to Reviewer AFg8**
> > >
> > > **Re1:** Thank you for your suggestion. As suggested by Reviewer AFg8, we experimented with regularization for discriminators of baselines. We ran experiments for 3 seeds in the Ant 0.03 noise setting on the two strongest learning from observation baselines: GAIfO-s and GAIfO. GAIfO-s without regularization achieves 18.22% success, with mixup regularization [1] 19.78% success, and with spectral normalization [2] 27.53% success, while our method achieves **48.14%** success. Likewise, GAIfO without regularization achieves 5.22% success, with mixup regularization 3.94% success, and with spectral normalization 5.73% success, much lower than the 48.14% success with our method.
> > >
> > > In summary, while regularization can help prevent baselines from overfitting their discriminators as pointed out by the reviewer, it cannot fully resolve the overfitting issue and therefore the baselines still achieve worse generalization performance than our method. Note that we do not try regularizing the discriminator for SQIL since it is not an adversarial imitation method and thus it does not have a discriminator. We will add this result in the revised version.
> > >
> > > **Re2:** Yes, we agree with your statement, except for the phrase “...with the former [absolute reward] encouraging solving the task more quickly”. The absolute reward encourages the agent to solve the task *slower* to accumulate more reward since the proximity value is positive. We agree this explains the difference for tasks like Ant Reach or Hand Rotate through the positive reward helping the agent not terminate. We will revise the paper to incorporate this discussion.
> > >
> > >
> > > [1] Zhang et al. "mixup: Beyond empirical risk minimization." ICLR 2018
> > >
> > > [2] Miyato et al. "Spectral normalization for generative adversarial networks." ICLR 2018

---

### Official Review · Reviewer_RPMC · 2021-07-18

**Rating:** 6
**Confidence:** 4

**Summary:**

The paper presents a learning from observation approach that uses a temporal goal proximity estimate as reward function for policy learning. The temporal goal proximity function is learned from observations and measures how many steps are left to reach the target goal state. Additionally, an ensemble of proximity functions is used to empirically estimate the uncertainty function that is added to the reward function to discourage exploration of uncertain proximity states. Results on a number of simulated tasks show performance improvement over GAIL, goal-GAIL, GAIfO, BCO and other competing methods.

**Limitations And Societal Impact:**

see above. no negative societal impact.

**Main Review:**

The paper is well-written, experiments are clean and well-presented, the ablation study helps understand the role of how different components of reward function are interacting. Results are also convincing on the presented tasks.

My main concern is that the task progress indicator seems to be a limiting signal to learn from expert observations. Although the heuristic is simple and works well on the example tasks, its usability may well be limited to low-dimensional and simple short-horizon tasks only. There needs to be some semantic mapping of the goal state and the current state (possibly in a latent space) besides temporal steps to reach the goal state. This would likely be more evident in long-horizon tasks and/or high-dimensional tasks.

- Limitation of temporal goal proximity function would also be evident if the expert demonstrations are sampled from optimal policy roll outs of different reward functions. The uncertainty estimate would likely be higher in such situations, even if the observations lie within the feasible set.

- It would be useful to describe what domain randomization (objects, backgrounds) are performed in the experiments, as the mapping of state to temporal progress should remain invariant.

- Results in Fig. 6(b) indicate that the offline imitation scenario is sub-optimal and online interactions are required for task completion. It would be useful to comment on the sample and time complexity in offline and online scenarios.

**Time Spent Reviewing:**

3

---

> ### Author Response · Authors · 2021-08-10
> **Response to Reviewer RPMC**
>
> We thank the reviewer for the constructive feedback and address the concerns in detail below.
>
>
> **1. The proposed method may be limited to low-dimensional and short-horizon tasks.**
>
> We argue that our goal proximity can provide a sufficient learning signal for high-dimensional and long-horizon tasks.
>
> Firstly, our goal proximity measures temporal progress (not spatial distance); therefore, this single scalar entails all _semantic information_ about how to reach the goal from a state, as explained in L123-125. Therefore, our goal proximity reward provides a semantically meaningful learning signal from high-dimensional inputs. For example, the visualization in Fig. 4d demonstrates that we learn a goal proximity model based on the semantic, non-euclidean relationship between high-dimensional observations and goals.
>
> Moreover, our experiments on relatively high-dimensional observations (Navigation and Hand Rotate) show some potential of our method for such complicated tasks. Tackling high-dimensional and long-horizon tasks is challenging not just for our method, but is also a  fundamental challenge for IL and RL, due to difficult representation learning, exploration, and temporal credit assignment. We believe our method can be further improved to handle such complex tasks by integrating recent advances in generalizable and efficient IL and RL techniques, which are complementary to our method.
>
> We will make these points clearer in the revised manuscript.
>
>
> **2. The proposed goal proximity function would not handle demonstrations from different reward functions.**
>
> In this paper, we aim to tackle _goal-directed_ imitation learning, where all demonstrations are optimal under the same goal-reaching reward. Leveraging (task-agnostic) demonstrations obtained from different reward functions is a promising research direction to explore [1-5]. However, we believe this is out of scope for this work. Specifically, one potential way is to replace the GAIL reward with our generalizable goal proximity reward in skill-prior imitation learning [5].
>
>
> **3. Describe what domain randomizations are performed in the experiments.**
>
> In our experiments, we randomized the starting agent pose and velocity, object locations, and goal (Appendix F.1). We demonstrated in Fig. 3 that prior methods are not robust to these randomizations. Since the proximity function must model precise, continuous-valued information about task progress (goal proximity), it is expected to capture task-relevant information and be robust to further randomizations. We will mention this in the main paper.
>
>
> **4. Sample and time complexity in offline and online goal proximity update in Fig. 6b.**
>
> The _offline scenario_ in Fig. 6b does not refer to offline _policy_ training but offline _goal proximity function_ training. Thus, both scenarios require the same amount of online interactions to train a _policy_ that maximizes the goal proximity reward. But, we use this online experience to train both the policy and goal proximity function in the online scenario, while we update only the policy in the offline scenario. In summary, there is no significant difference in terms of sample and time complexity. Therefore, the samples needed to train the policy are also “free” supervision for the proximity function.
>
> Thank you for pointing out this confusion. We will clarify the definition of online and offline training of the proximity function in the revised version.
>
> **References**
>
> [1] Lynch et al. “Learning Latent Plans from Play”, CoRL 2019
>
> [2] Pertsch et al. “Accelerating Reinforcement Learning with Learned Skill Priors”, CoRL 2020
>
> [3] Ajay et al. “OPAL: Offline Primitive Discovery for Accelerating Offline Reinforcement Learning”, ICLR 2021
>
> [4] Singh et al. “Parrot: Data-Driven Behavioral Priors for Reinforcement Learning”, ICLR 2021
>
> [5] Pertsch et al. “Demonstration-Guided Reinforcement Learning with Learned Skills”, arXiv 2021
>
> &nbsp;
>
> Please kindly let us know if there are any further concerns or missing experimental results that potentially prevent you from accepting this submission. We would be more than happy to address them. Thank you very much for all your detailed feedback and the time you put into helping us to improve our submission.

---

> > ### Author Response · Authors · 2021-08-28
> > **Re: Response to Reviewer RPMC**
> >
> > We thank the reviewer for the thorough and constructive initial review.
> >
> > As the end of the discussion period is approaching, please kindly let us know if there are any clarifications that we can make or experiments that we can do. We would be more than happy to address them if time allows.
> >
> > Thank you very much for all your detailed feedback and the time you put into helping us to improve our submission.

---

> > ### Comment · Reviewer_RPMC · 2021-08-31
> > **response to authors**
> >
> > Thanks for the response. I agree with most of the feedback !
> >
> > - My concern remains around situations where the temporal progress indicator may not be a unique signal for each state. In case of such temporal progress variations in the demonstrations (and if the state demonstrations may not vary in a similar way), it is not clear to me that if the learned behavior would replicate the expert demonstrations.
> >
> > - Would it be fair to say that minimum-time problems may be well-suited for the proposed approach ? I would further appreciate if the authors can describe a limitations section where the approach intuitively may not generalize from demonstrations.

---

> > > ### Author Response · Authors · 2021-08-31
> > > **Response to Reviewer RPMC**
> > >
> > > We thank the reviewer for the response and are happy to see that our response addressed the concerns in the initial review. Here is our response to your follow-up questions:
> > >
> > > * In this paper, we assume the demonstrations are optimal and the goal proximity only depends on the temporal distance to the goal, which ensures that demonstrations for the same goal are similar. Thus, the goal proximity function can stably provide task progress to train an agent.
> > >
> > > * As Reviewer RPMC mentioned, our approach is well-suited for (goal-directed) minimum-time problems as we learn temporal task progress from optimal demonstrations. Please note that our method is designed for learning general goal-directed tasks and our goal proximity function can entail all information about the task, goal, and any roadblocks (L123-125).
> > >
> > > * The limitations of our method in generalization to unseen environments and different embodiments were discussed in the paper (L39-41, L347-350) and in [our response to Reviewer YZmg](https://openreview.net/forum?id=lp9foO8AFoD&noteId=wBJaQRZ2ax5). As Reviewer RPMC mentioned above, handling large temporal progress variations and suboptimal demonstrations can be a promising future work to further extend our approach. As per the reviewers’ request, we will elaborate more on these limitations in the revised paper.
> > >
> > > Thank you again for your follow-up. Please let us know if any further concerns potentially prevent you from accepting this submission.

---

> > > > ### Comment · Reviewer_RPMC · 2021-08-31
> > > > **response to authors**
> > > >
> > > > Thank you for further clarification! I have updated the score.

---

> > > > > ### Author Response · Authors · 2021-09-01
> > > > > **Re: response to authors**
> > > > >
> > > > > We thank the reviewer for providing valuable feedback and recognizing our responses.

---

### Official Review · Reviewer_W3nz · 2021-08-03

**Rating:** 7
**Confidence:** 4

**Summary:**

The paper addresses the problem of learning from observation for directed tasks with a focus on generalization beyond the demonstrations. The main contribution of the paper is a goal proximity function (also referred as goal progress in the paper) which predicts the agent progress towards the goal. The paper has extensive experiments on navigation, locomotion and robotic manipulation which proves that the proposed goal proximity function helps to generalize in imitation learning. The results show that proposed method achieves better performance in case of learning from observation tasks and comparable performance with learning from demonstrations method.

**Limitations And Societal Impact:**

- for societal impact, the authors mention about the method increases "the ability of the autonomous agents", Can you specify what types of ability will be improved or mention specific ability for autonomous agents that will be improved with the proposed method?
-  Given that goal proximity is a simple measure of steps to the goal from current state, I am having difficulty in understanding that how the proposed method with help in generalization of states and task? The author talks about generalization in the future work, could the authors elaborate it? If space is limited then authors are encouraged talk/discuss about generalization in a separate section in the Appendix.
-What kind of new goals the method can generalize to?

**Main Review:**

- The paper is well-written and easy to follow.
- The proposed method is evaluated on good range of experiments
- The framework of learning the proximity function from expert demos and agent experience are nice contributions.
- good comparison with other related methods (GAIL, Goal GAIL etc.,)

**Time Spent Reviewing:**

8

---

> ### Author Response · Authors · 2021-08-10
> **Response to Reviewer W3nz**
>
> We thank the reviewer for the constructive feedback and address the concerns in detail below.
>
>
> **1. How does “task progress” or “goal proximity” help generalization?**
>
> Our method helps generalize to new goals and states by estimating the goal proximity which encourages the use of task-relevant features. Precisely predicting goal proximity on a _continuous_ scale requires learning details of which states are closer to the goal, an important detail for goal-directed tasks. This is opposed to prior work which discriminates agent and expert behaviors by predicting _binary_ labels, which is prone to overfitting to task-irrelevant features. Our paper elaborated on this generalization further (L30-36 and L132-137), and we will make this information more visible in the revised version.
>
>
> **2. What kind of new goals can the method generalize to?**
>
> Our method can generalize to regions of the goal space not covered by the expert and larger perturbation in goal sampling (L175-183). For instance, in Navigation, the expert demonstrations only cover 25% of goal states as visualized in Fig. 11 in appendix, but the agent must reach goals not present in the expert demonstrations. We will make this clearer in the revised version.
>
>
> **3. More in-depth discussion regarding generalization described in the future work.**
>
> As described in conclusion, the _generalization_ ability in imitation learning can include generalization to (1) unseen states and goals, (2) new visual environments (e.g. background), (3) unseen objects, and (4) different embodiments (e.g. humans to robots or different dynamics).
>
> In this paper, we focus on generalization to (1) unseen states and goals. This is especially important in imitation learning when the number of demonstrations is not sufficient to cover all possible states and goals. This is very common in imitation learning due to costly demonstration collection. Our approach suggests an effective way of using the demonstrations with limited coverage by learning the generalizable goal proximity reward.
>
> We agree that generalization to a different environment and embodiment is another important research direction and this is indeed our immediate future work. As Reviewer YZMg mentioned, recent advances in generalizable representation learning [1-3], robust policy learning [4-5], and cross-domain correspondence [6-7] enable us to train a policy that generalizes to new environments and embodiments. Note that these approaches are orthogonal and complementary to our method as our goal proximity function can be trained on top of the learned representations [1-7]. We believe that our method can be combined with these approaches and improve their performance with better demonstration efficiency and additional supervision about task progress. We will include this discussion in the revised version.
>
>
> **4. Specify what types of ability of autonomous agents will be improved with the proposed method (in Societal Impact)?**
>
> In this paper, we focus on the ability of imitating experts (e.g. humans) from observation. This enables autonomous agents to utilize data even without expert actions, such as kinesthetic demonstrations and video demonstrations. Ultimately, it could allow autonomous agents to acquire skills even from watching Youtube videos.
>
> **References**
>
> [1] Sermanet et al. “Time-Contrastive Networks: Self-Supervised Learning from Video”, ICRA 2018
>
> [2] Srinivas et al. “Universal Planning Networks”, ICML 2018
>
> [3] Wang et al. “NerveNet: Learning Structured Policy with Graph Neural Networks”, ICLR 2018
>
> [4] Kumar et al. “RMA: Rapid Motor Adaptation for Legged Robots”, RSS 2021
>
> [5] James et al. “Transferring End-to-End Visuomotor Control from Simulation to Real World for a Multi-Stage Task”, CoRL 2017
>
> [6] Zhang et al. “Learning Cross-Domain Correspondence for Control with Dynamics Cycle-Consistency”, ICLR 2021
>
> [7] Zhang et al. “Policy Transfer across Visual and Dynamics Domain Gaps via Iterative Grounding”, RSS 2021

---

> > ### Author Response · Authors · 2021-08-28
> > **Re: Response to Reviewer W3nz**
> >
> > We thank the reviewer for the thorough and constructive initial review.
> >
> > As the end of the discussion period is approaching, please kindly let us know if there are any clarifications that we can make or experiments that we can do. We would be more than happy to address them if time allows.
> >
> > Thank you very much for all your detailed feedback and the time you put into helping us to improve our submission.

---

### Decision · Program_Chairs · 2021-09-27

**Decision:**

Accept (Poster)

**Comment:**

The author response convinced all reviewers, they now unanimously vote for an accept.